# Adaptive Sparse Federated Learning in Large Output Spaces via Hashing

## Abstract

This paper focuses on the on-device training efficiency of federated learning (FL), and demonstrates it is feasible to exploit sparsity in the client to save both computation and memory for deep neural networks with large output space. To this end, we propose a sparse FL scheme using hash-based adaptive sampling algorithm. In this scheme, the server maintains neurons in hash tables. Each client looks up a subset of neurons from the hash table in the server and performs training. With the locality-sensitive hash functions, this scheme could provide valuable negative class neurons with respect to the client data. Moreover, the cheap operations in hashing incur low computation overhead in the sampling. In our empirical evaluation, we show that our approach can save up to $70\%$ on-device computation and memory during FL while maintaining the same accuracy. Moreover, we demonstrate that we could use the savings in the output layer to increase the model capacity and obtain better accuracy with a fixed hardware budget.

## 1 Introduction

Recently, federated learning (FL) [18] and its applications [36, 14, 17, 23] receive attentions from both research community and industry. FL defines a practical yet challenging task: given a set of devices where each device maintains its private data locally, we would like to collaboratively train a model on these devices without data exchange. Significant effort has been made in improving optimization strategy [15, 31], privacy protection [8] and fairness [16] of FL.

**A Challenge in On-device Training:** FL introduces a device shift in the distributed training of machine learning models. In the cloud center, we were able to train large-scale foundation models on massive graphic processing units (GPUs) in a centralized way. In FL setting, our training hardware is limited to system-on-chip (SoC) on mobile devices. This shift leads to significant efficiency issues in FL: (1) The models we trained on GPU clusters are giant in terms of parameters. It is standard to have billion-scale parameters for language [2] and recommendation [19] models. However, the memory constraint for FL devices forces us to limit the model parameter size to make on-device training feasible, which causes a model size mismatch between FL and centralized models. This mismatch would degrade the FL model performance and prevent us from the benefits of large deep models. (2) In the centralized training, the advantages of specialized hardware such as TPU provide efficient matrix multiplication for training deep neural networks. However, the on-device tensor chips like TPUs in Pixel phones [24] are not as powerful as their serverside counterpart, which would significantly improve the training efficiency in FL. (3) The training of deep models through forward and backward propagation requires memory to store the intermediate results. Same the previous two issues, the hardware constraint of mobile devices would affect the efficiency of memory read and write during training, which exaggerates the computation overhead in FL.

**An Opportunity of Sparsity:** There is an emerging trend on exploring sparsity in neural network training [27, 7, 6, 10]. An interesting direction is to switch the matrix multiplication from dense to sparse mode. For instance, given an input matrix $X \in \mathbb{R}^{n \times d}$ and the linear layer weight matrix $W \in \mathbb{R}^{d \times m}$, we view each row of $X$ as an embedding and each column of $W$ as a neuron. In this way, for each embedding in $X$, we could only select a subset of neurons in $W$ for computation. As a result, we perform a sparse version of operation $XW$. Well-known research literature in this area includes the lottery ticket hypothesis [12, 37], sub-linear deep learning engine (SLIDE) [7, 6] and independent subnet training [38]. In this paper, we argue that there is an opportunity to improve the on-device training efficiency of FL with sparsity. Firstly, although the sparse training strategy does not change the model architecture, it only activates a subset of model parameters in each iteration. This feature could help us in FL so that each client only selects a subset of trainable parameters from the model for iterative optimization. As a result, the on-device parameter size would be reduced. Moreover, the sparse alternative to the matrix multiplication could significantly reduce the computation overhead, making FL easy on CPU only devices. Furthermore, the sparse training generates sparse intermediate results, which also reduces the memory access on the device.

**Exploiting Adaptive Sparsity in Large Output Spaces:** In this paper, we focus on the sparse FL for deep neural networks in large output spaces (LOS). LOS is common in deployed deep models. For instance, in language processing tasks such as next word prediction [20] and question answering [21], the output space would be the vocabulary size. In recommendation systems [19], the output space would be the number of products in the database. In both cases, the number of classes in the output space could be enormous. As a result, the output linear layer would contain the most model parameters if we would like to train them on-device using FL. On the other hand, since we would perform Softmax function on the output logits, we could approximate the output layer by focusing on the logits with high values. In fact, the LOS would be a perfect scenario for sparse training. If we could adaptively pre-select the neurons in the output linear layer that may incur a large inner product with the hidden input vector, we could only do forward and backward computation on the selected neurons [1, 7, 10, 6, 30]. Therefore, we could perform efficient on-device FL by saving both the computation and memory.

However, the combination of sparse training with FL could still be challenging: (1) An efficient design is required for sparse training in FL so that we can maintain the full model on the server and a sparse model on the client device. (2) It remains unknown whether the adaptive sparsity would be effective in the federated optimization with non-i.i.d data distribution. (3) How to use the saved computation and memory by sparsity for further improvements in the model accuracy. In other words, how to improve the model performance with a fixed hardware budget using sparsity?

## 1.1 Our Contributions

In this paper, we introduce an empirical study on the sparse FL in LOS. We propose to use hashing algorithms that adaptively select neurons in the output layer for forward and backward computation. Specifically, our contributions could be summarized as:

1. We introduce an adaptive hash-based sparse FL scheme for training in LOS. In this scheme, the server hashes the output layer's neurons in hash table. Next, the server sends the hash function to each client. Each client uses the hash function to generate hash codes of their own data. Next, each client uses the hash codes to look up the near neighbor neurons of its data from the server. Finally, each client only performs forward and backward propagation on the selected neurons.

2. We empirically show that the hash-based sparse training in the output layer is effective in the federated optimization. We could maintain the same model accuracy with $30\%$ parameters in the output layer. As a result, the on-device training efficiency would be improved.

3. We demonstrate that in the proposed hash-based sparse FL scheme, the saved on-device model parameters in the output layer would be used to improve the model capacity. We show that on a fixed on-device parameter budget, if we perform sparse training on the output layer and use the saved parameters to increase the embedding and hidden dimension of the model, we could have better accuracy with on-device FL.

## 2 Related Work

**Hashing Algorithms for Sparse Machine Learning:** The hashing algorithms have demonstrated empirical effectiveness in the sparse training of machine learning models [7, 10, 6, 34, 32]. [7] proposes SLIDE algorithm that uses locality-sensitive hashing (LSH) [11] to preprocess the neurons of a wide output layer in hash tables. Next, given a batch of embeddings, SLIDE use them as a query and lookup the neurons that are close in cosine similarity from hash tables. Finally, SLIDE only performs forward and backward computation in the selected neurons. [10] shows that SLIDE can be further accelerated with the advance in hardware. [6] demonstrates that SLIDE could be improved by learnable hash functions. [34] focuses on the Frank-Wolfe optimization algorithm. In particular, [34] preprocess the vertices of the weight space in hash tables. Next, given the current weight, instead of computing it with all vertices, we only need to compute with near neighbor vertices and choose one as the next direction. [32] proposes parallel memory writing algorithms for the sparsified gradients in the data-parallel distributed training and provide up to $3.52\times$ speedups.

**Efficient On-device Training in FL:** There are two major techniques for improving the on-device efficiency of FL. The first technique is named partial variable training (PVT) [35, 26]. PVT aims at freezing a fraction of trainable parameters when we train models on the client devices. For instance, we could freeze some of the fully-connected layers and only perform federated optimization on the other parts of the model. [26] focuses on saving the communication cost of transferring model gradients. [35] is also trying to save the on-device model size and memory for intermediate results. Meanwhile, we observe some computation saving in the backpropagation. Another technique is called federated dropout (FedDrop) [3, 9]. The FedDrop randomly selects a subset of neurons from the model and sends it to the client for training. Although FedDrop saves the on-device computation and memory, the nature of randomness would cause an accuracy gap between the FedDrop and original training when we increase the sparsity. The major reason behind this phenomenon is that FedDrop's random sampling is not adaptive to the status of input embeddings. For instance, it is shown that the neurons with large inner products to input embedding should be a more important example in the output layer. As a result, the missing of these neurons would lead to slower convergence.

## 3 Method

In this section, we introduce our hashing algorithms for sparse federated learning in wide output layer. We start with a formal introduction of hash-based sampling. Next, we propose a sparse FL scheme using hashing.

### 3.1 Hash-based Sampling in Neural Network

In this section, we present how to use hash-based Sampling [27, 33, 7] in the training of neural network. We start with introducing a simple yet effective LSH function, namely SimHash [4].

**Definition 1** (SimHash). *Let $K$ denote the number of hash bits. Let $A \in \mathbb{R}^{K \times d}$ denote a random matrix where each entry is drawn i.i.d from normal distribution $\mathcal{N}(0, 1)$. Given an input vector $x \in \mathbb{R}^d$, we define the SimHash function $h : \mathbb{R}^d \to \mathbb{R}^k$ as*

$$h(x) = \textsf{sign}(Ax),$$

*where $\textsf{sign}$ is an element-wise sign function that set nonzero values to 1 and other values to 0. Moreover, we show that for any two vectors $x, y \in \mathbb{R}^d$,*

$$\Pr[h(x) = h(y)] = (1 - \frac{\theta_{xy}}{\pi})^K,$$

*where $\theta_{xy}$ is the angle between $x$ and $y$.*

The SimHash's definition (see Definition 1) suggests that if two vectors are close in angle, with high probability they would have the same hash code. Moreover, previous work suggests that with a pair of asymmetric transforms applied on $x$ and $y$ [25], respectively, the collision probability $\Pr[h(x) = h(y)]$ would be monotonic to the inner product $x^\top y$. Taking advantages of this property, the hash-based sampling algorithm for a linear layer can be summarized as below:

1. Given a weight matrix $W \in \mathbb{R}^{d \times m}$, extract each column $W_i$ from $w$ and compute $h(W_i)$. Build a hash table that allocates $W_i$s with the same hash code in a bucket.

2. Given a batch of embedding $X \in \mathbb{R}^{n \times d}$, for each row $x_j$ in $X$, compute $h(x_j)$ and lookup the $W_i$s that has the same hash code with $h(x_j)$. Next, we take the union of the retrieved $W_i$s and write it as matrix $W_{\text{select}}$. Finally, we subsample on $W_{\text{select}}$ with fix sparsity ratio and compute $XW_{\text{select}}$ for forward/backward propagation.

In practice, we use $L$ hash tables and take the union of weight columns retrieved by each hash table. Noted that both $K$ and $L$ are tuning parameters. For an input batch of embedding $X$, hash-based sampling could return a sub-matrix $W_{\text{select}}$ that the inner product between each row in $X$ and each column in $W_{\text{select}}$ may incur large inner product. In LOS, $W_{\text{select}}$ represents the neurons that may have larger logits with $X$. In the practical setting, we may sub-sample on the columns of $W_{\text{select}}$ or randomly add more columns to $W_{\text{select}}$ so that we can fix the sparsity budget.

## 3.2 A Scheme of Sparse FL

In this section, we design a server-client scheme for sparse FL in LOS. As introduced in Section 3.1, hash-based sampling could select the neurons that may have large logits in the LOS. However, the maintenance of hash tables and the preprocessing of neurons may generate extra computation overhead. In fact, the computation in hashing the neurons should be carefully handled by smart scheduler in centralized training [6]. In our work, we take a FL view of this procedure and argue that the computation overhead can be reduced by the powerful computation resources in the server. Specifically, we introduce our scheme as below:

1. **Hash table maintenance:** The server uses SimHash (see Definition 1) and preprocesses the columns of the weight $W$ in the output layer in hash table. The server refreshes the hash table after each round.

2. **Client initialization:** When a new client comes, the server sends the hash function and the weights of layers before the output layer to the client. The client performs forward pass and generate embedding vectors of its own data.

3. **Client hashing:** The client hashes the input embedding using of data samples the received hash function and generates a set of bucket locations in the hash table.

4. **Neuron retrieval:** The client transfers the bucket locations to the server and looks up the neurons of output layer in the corresponding buckets.

5. **On-deivce training:** The client receives the lookup-ed neurons and performs forward and backward computation.

6. **Aggregation:** The client passes the model updates such as gradients back to the server for aggregation.

The advantages of the proposed scheme can be summarized as: (1) the client device does not have to maintain the hash tables, which reduces the computation overhead in preprocessing neurons during the centralized training, (2) the client only lookups a subset of parameters in the output layer, which reduces the communication cost and on-device model memory, (3) the client trains on neurons that may have higher logits in the output layer, which served as an effective negative sampling for faster convergence. It is normal that the number of neurons retrieved from hash tables is larger than the client budget. We can compute the activations of these neurons with the client data and only keep the large activation neurons on-device for backpropagation. Note that our approach may involve more communication between servers and clients. We could use the FedSelect [5] to look up neurons with privacy protection.

## 3.3 Improving Model Capacity in Fixed Budget

As shown in the previous sections, our sparse FL scheme with hashing reduces the on-device parameter size for the last output layer. For a SoC with fixed hardware memory budget, we could use the saved space to increase the trainable parameters in other parts of the model to have better performance. We suggest two major directions: (1) increase the hidden dimension in both embedding and linear layer for better feature representation, (2) add more attention blocks [29] or linaer layers for better feature mixing. In the experiment section, we will discuss how these two directions would help us improve the empirical performance of on-device FL.

Table 1: Parameter Size of Transformer Model for Stackoverflow Dataset. We also include the percentage of token embedding (ouput layer) in the model.

| Emb. Dim | FFN. Dim | Attn. Size | Emb. Size (10K) | Emb. Size (80K) |
|----------|----------|------------|-----------------|-----------------|
| 96 | 1536 | 330K | 960K (49.2%) | 7.68M (88.5%) |

# 4 Experiment

In this section, we introduce an empirical evaluation of the proposed sparse FL training scheme in LOS with hashing. We start with introducing the next word prediction task we focus on. Next, we introduce the models we evaluate. Finally, we present the experimental results with an abalation study. We also provide a visualization of hash tables in Appendix A.

## 4.1 Settings

**Dataset.** We evaluate the proposed sparse FL scheme on the next word prediction task using Stack Overflow (SO) dataset. The SO dataset contains 342477 training clients. The total training example size is 135M. The SO dataset has 38758 clients for validation with dataset size 16M In the next word prediction setting, we take the first 256 sentences and truncate each sentence to a sequence length 20. We aim to predict the next word given the previous context words. It is standard to set the vocabulary size to 10K by taking the most frequent words from the training data. In our paper, we also extend the vocabulary size to 80K since larger vocabulary has larger word coverage.

**The Transformer Model.** In this paper, we study the performance of Transformer model. We profile the parameter size of transformer models in Table 1. In the model, we set the token embedding size as 96. We also set the hidden dimension of the Q, K, V layer in attention as 96. For the FFN dimension in attention block, we set it to 1536. There would be 330K parameters for each attention block. But the embedding table size would be 960K for 10K vocabulary and 7.68M for 80K vocabulary. In this paper, we shared the weights between the embedding and the last output layer so that the largest parameter tensor in the model is the embedding table. In this case, if we train the full model on each client, we have to send all the embedding tables to the client so that they can use them for output layer. On the contrary, if we could perform sparse training on the output layer by selecting a subset of neurons, we only need to send a subset of embeddings from the embedding table to the client. It is obvious that this scheme would save the transmission of the largest weight tensor.

**Parameters.** In the federated optimization, we use SGD as the client optimizer and Adam as the server optimizer following FedAdam approach introduced in [22]. For the Adam optimizer in the server, we vary the epsilon between $10^{-3}$ and $10^{-4}$. We also perform a grid search on server learning rate set $\{10^{-1}, 10^{-1.5}, 10^{-2}\}$ and client learning rate set $\{10^{-1}, 10^{-1.5}, 10^{-2}\}$. The training steps and rounds follow the same setting as [31]. We chose the $K$ and $L$ shown in Section 3.1 from $\{2, 4, 6, 8\}$. We vary random seeds in both model initialization and data loader for a fair comparison. For the evaluation metrics, we use the accuracy with the out-of-vocabulary, padding, EOS and BOS tokens masked.

## 4.2 Results in Sparse FL

To start with, we would like to evaluate the performance of our hash-based sampling in the output layer. Specifically, we would like to answer the following question: does hash-based sampling achieves better accuracy than random sampling in the training in LOS with different sparsity? Here we conduct an experiment on the Stack Overflow dataset, we vary the sparsity level and compare the hash-based sampling with random sampling. In Figure 1, we present the evaluation accuracy versus the actual computed parameters. For each parameter, we repeat the experiment for 3 times and plot the average accuracy. As shown in the figure, if we fix the computed parameter, hash-based sampling is able to outperform random sampling with better final accuracy. Moreover, we show that, with above 30% of the parameters, the hash-based sampling is able to be less than 0.05% full training accuracy. Noted that the model large vocabulary size can predict more out-of-vocabulary words. In Figure 1 we do not use a unified evaluation accuracy across different vocabulary. But we do observe

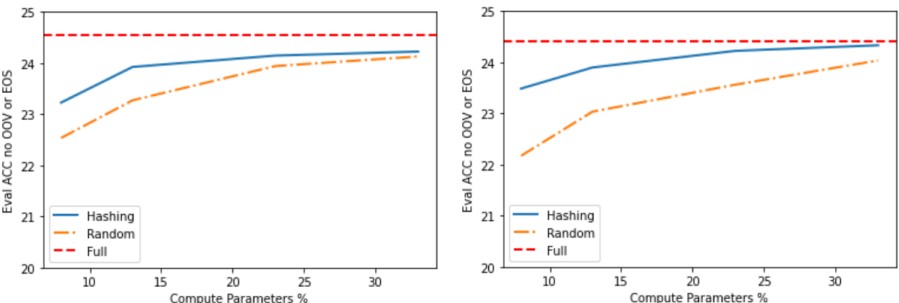

Figure 1: Evaluation accuracy versus the computed parameters in the output layer. Here the percentage of the computed parameter represents the sparsity level in the training. Left: vocabulary 10K, Right: vocabulary 80K. Note that the red line indicates the training accuracy if we compute on all the parameters in the output layer.

Table 2: Model accuracy in fix parameter budget. Here Params. represent the total number of parameters for the model. Vocab. represents the vocabulary size.

| Emb. Dim | FFN. Dim | LOS Parameters | Sparse Approach | Vocab. | Params.(M) | Eval. Acc. |
|----------|----------|----------------|-----------------|--------|------------|------------|
| 96 | 1536 | 100% | Full | 10K | 1.92 | 24.54±0.16 |
| 96 | 2560 | 30% | Hashing | 10K | 1.71 | **24.72**±0.16 |
| 96 | 1536 | 100% | Full | 80K | 8.65 | 24.41±0.10 |
| 96 | 8192 | 30% | Hashing | 80K | 6.34 | **25.20**±0.09 |

that transformer with 80K vocabulary has better accuracy in the unified evaluation accuracy. In the next section, we would like to show how to use this observation for better on-device training accuracy.

### 4.3 Model Improvement in Fixed Budget

In this section, we study how to improve the model accuracy with a fix parameter. Specifically, we would like to answer the following question: does the on-device parameters saving in the output layer help us improve the model by adding parameters in other parts? Here we use the parameter size of the full transformer model described in Section 4.1 as our budget. We would like to apply sparse training in the output layer while increasing the embedding dimension so that we could get closer but not exceed the parameter size budget. Moreover, with the knowledge from Section 4.2, we only keep 30% of the parameters in the output layer. In Table 2, we present the results. For vocabulary size 10K, if we use hash-based sampling with 30% compute parameters in the output layer and increase the FFN dimension to 2560, we can outperform the original full model training accuracy in evaluation dataset. Moreover, if we increase the vocabulary size to 80K, the improvement of hash-based sampling would be enlarged. If we only select 30% parameters in the output layer using hash-based sampling and increase the FFN dimension of Transformer to 8192, we could maintain a lower parameter size on device. Moreover, we could significantly improve the evaluation accuracy to 25.2%. increase the FFN dimension of Transformer to 8192, These experiments validate that our hash-based sparse FL scheme is able to improve the model performance without increasing the on-device parameter size.

## 5 Discussion

In this section, we would like to discuss our observations and the potential future directions of this work. Firstly, we observe that for smaller vocabulary sizes, LSH performs marginally better than random sampling. With our analysis, we observe that LSH does not retrieve large inner product neurons with high recall. Meanwhile, the exact maximum inner product search (MIPS) on neurons gives us better accuracy. In this case, a promising future direction would be the introduction of more MIPS data structures such as quantization [13] and proximity graphs [39, 28] . Secondly, we would like to explore the opportunity of further increasing the vocabulary size for the on-device language modeling. Our experimental results have suggested that LSH approach performs better as we increase the vocabulary. We would like to investigate how our approach overcomes the on-device hardware

limit. Thirdly, our approach requires communications of hash codes and neurons. We would like to combine this approach with more secure and efficient communication schemes [5] for aggregation.

## 6 Conclusion

In this paper, we propose a sparse federated learning (FL) scheme using a hash-based adaptive sampling algorithm. We argue that during the FL training of deep neural networks in large output space, we can sample a subset of neurons in the output layer and perform forward and backward propagation on these neurons only. Moreover, we introduce a hash-based adaptive sampling approach in the neuron sampling for FL. We pre-index the neurons of the output layer in hash tables. Next, given the input embedding to the output layer, we could look up its near neighbor neurons from hash tables for the sparse training. Furthermore, we introduce a sparse FL scheme based on this hash-based sampling approach. In our scheme, the server takes over the neuron indexing and maintains the hash tables, while the client only maintains a subset of neurons in the last layer through hash table lookups. In this way, we show via extensive experiments that we could use around 30% parameters in the last layer and obtain the same final accuracy as full parameter training. We also show that with our approach, we could perform sparse training in the output layer and use the saved parameters to improve the model capacity in embedding and fully-connected layers. This design leads us to better on-device FL accuracy with the same parameter budget.

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

# Appendix

## A  Visualization of the Hash Table

In this section, we visualize the bucket in the hash table we built for Stack Overflow dataset. We showcase the tokens of three buckets from a hash table as below. Here we set the $K = 4$ (see Section 3.1).

1. java, python, ruby, spring, tomcat, gcc, swift, println, opencv, openssl, activerecord, gdb, clang, webforms, rpc, i18n, nunit, llvm, msvc, apt, filenotfoundexception, openjdk, teardown, rejection

2. problems, ways, issues, questions, great, solutions, tutorials, major, bugs, conventions, viable, disadvantages, strategies, interactions, measures, useful, considerable, findings, documentations, sensors, reaction, brilliant, orthogonal

3. change, go, define, modify, ask, perform, manage, determine, accept, reset, combine, maintain, bring, evaluate, optimize, customize, jump, automate, preserve, terminate, associate, buy, pip, synchronize, eat, mutate, assemble, recalculate, optimise, dotnet, check-in

We observe that the first bucket is a programming language bucket. It contains "java", "python" and related platforms such as "opencv". In the second bucket, we observe that there are similar tokens such as "great", "brilliant" and "useful". Also the "issue" means similar to "questions" and "bugs". In the third bucket, there is a set of synonyms, such as "change", "modify" and "reset." To sum up, the hash table is able to group the relative tokens in the same bucket and we can look them up with a query.

