# OpenReview forum: "Adaptive Sparse Federated Learning in Large Output Spaces via Hashing"
_NeurIPS.cc/2022/Workshop/Federated_Learning — FL-NeurIPS 2022 Poster_

### Official Review · Reviewer_uZpG · 2022-10-17
**good idea but has a few limitations**

This paper proposed an approach to perform sparse training for FL  via hashing. The paper is well-written. The paper is worth reading and proposed an interesting idea. However it has two major limitations:

1. In the Sparse FL scheme, each client needs to double the communication with the server. In the first communication, the client computes the forward loop before the output layer, and sent the hash location to the server. The server then perform a second communication to the clients and sending the output neurons at the collided hash location.

2. While the experiments show that the proposed scheme outperforms random hashing at the same sparsity level and fixed parameter budget, the proposed method is only applicable to perform sparse training at the output layer, whereas random hashing can be performed at any layer. Since potentially random hashing can reach much higher sparsity.

---

### Official Review · Reviewer_HZ6U · 2022-10-17
**Large-Output-Space models can pose problems in FL scenarios since the output layer of a newtork can easily account for the majority of the model parameters. The Authors propose a framwork that delays the communication of the last layer until it is needed, reducing in this way communication and training costs.**

The Authors address the challenge that large-output-spaces (LOS) models pose in FL settings. This is an important aspect that probably has been overlooked by others in the community (including myself) since these models include the majority of their parameters in their output layers (which are tied to, for example, the number of output classes) and therefore do not offer the same flexibility in terms of optimizations that can be applied to them (e.g. pruning).

The Authors rely on a existing framework, SLIDE, and adapt it to the FL setting and in particular to LOS scenarios. In this way: (1) FL clients receive all the model parameters from the server except those in the final output layer; (2) clients perform standard forward propagation up until the last layer; (3) client hash the embeddings (that would be passed to the final layer eventually) using the hash functions received by the server; (4) clients communicate to the sever which weights they need for the final layer; and, (5) the clients receive the subset of weights that would be active in the last layer (therefore the last layer is sparse) and complete forward pass and the to standard backward pass.  Following this methodology the Authors obtain good results (i.e. close to that if the whole model was used by clients) while requiring <30% of the parameters in the last layer.

I like this paper but there is one important limitation: to complete each training step the clients need to communicate with the server to retrieve the weights for the output layer. This will severely slow down training since instead of completing a the remaining of the training (final layer forward + backward) would likely require less than a second, now it will likely require several times more as the idle training time for communicating with the server has to be taken into account. This aspect is not discussed in the paper, I'd recommend the Authors to study this in several scenarios as it is the main limitation of this work.

Following the topic above, the Authors could consider that, as a client completes several training steps, it is likely that over time the client as accumulated a large portion of the weights for the final layer (sent by the server over many training steps). This could lead to an interesting study and, potentially be a way to amortise the communication costs during training in the early stages of on-device training. However, this would only be possible if clients do require training for many steps. An interesting experiment would be how many training steps are required for this method to be worth it given a model X, a communication link with bandwidth Y and clients with compute/memory capabilities Z.

Other minor comments:
*    I think having an introduction split into paragraphs makes it a bit odd to read because, at least in the way it was written, it feels very much like a related work / background section.
*    I would advice the Authors to include a small diagram showing how SLIDE works in a client, otherwise it's hard to understand the basics of SLIDE for those unfamiliar with it. For example, having a model in a client, showing that it does forward until the last layer and then the hashing happens, communicates with server, receives weights and client completes training step. I suspect Figure 3 in the SLIDE paper could be useful to illustrate some of this.
*    It would be interesting to see other LOS models/dataset being evaluated.
*    Table 2 is exceeding the right marging

---

### Official Review · Reviewer_79dJ · 2022-10-18
**This paper propose a sparse federated learning (FL) scheme which utilize the hash-based sampling method to achieve the sparsification of the model in the federated environments. However, technical novelty and experimental evaluation are both limited.**

Pros:
1. This paper focus on the sparse FL for deep neural networks in large output spaces (LOS) and only does forward and backward computation on the selected neurons.
2. The experimental results show that it can maintain the same model accuracy with 30% parameters in the output layer.

Cons:

1. Lack of comparisons with other baseline sparsification models. Lack of comparisons on communication and computation costs.

2. The technical route is relative simple and straightforward.  More discussion on the choice of the hashing-based scheme should be included to justify the technical approach.

3.Since this method only sparses the output layer, the actual savings on the parameters are traded-off with the calculation and communication costs caused by the maintenance of the hash table and hashing functions.  Therefore, it is advised to quantify the actual benefits by taking into account these costs.

---

### Decision · Program_Chairs · 2022-10-20

Accept (Poster)